# COVID-19 pandemic preparedness period through healthcare workers' eyes: A qualitative study from a Romanian healthcare facility

**Laura Elena Stoichitoiu** [1,2] *, **Cristian Baicus** [1,2]

**1** University of Medicine and Pharmacy "Carol Davila", Bucharest, Romania, **2** Department of Internal Medicine, Colentina Hospital, Bucharest, Romania

* laura.elena.stoich@gmail.com

## Abstract

### Introduction

Healthcare providers represent a limited resource, and their mental health is crucial for patient care and for ensuring containment of the pandemics. We aimed to explore how healthcare workers experienced the preparedness period of COVID-19 pandemic, in order to ascertain the perceived weaknesses and strengths.

### Methods

Interviews were conducted with 17 participants encompassing senior physicians, residents, and nurses. They were audio-recorded, and the transcription was verbatim. We used thematic analysis.

### Results

We identified four themes, with subsequent subthemes: dealing with the unknown, human versus doctors, sense of helplessness, and a bridge to heaven, which explore how healthcare workers experienced the lack of knowledge, their feeling of losing control, and how they managed their internal fights. The disappointment provoked by the authorities and their colleagues was further evaluated. We identified factors involved in their well-being.

### Conclusions

COVID-19 pandemic represented and will still pose a challenge for healthcare workers (HCWs) from all over the world. They felt unprepared for such a crisis. Further measures should be implemented in every hospital to maintain HCWs awareness and to prevent physical imbalance. Appropriate standards of care should be further stated by the authorities so that the healthcare providers may find easier a balance between their safety and their patients' needs. Conducting qualitative research involving HCWs during pandemic times may help in informing more significant policy decisions.

**Data Availability Statement:** Data cannot be shared publicly because the informed consent stipulated that the transcripts will be shared only with the team members. Therefore, the Colentina Hospital Ethics Committee of Research considers

that we do not have the right to make all data underlying the findings described fully available. Data are available from the Colentina Hospital Ethics Committee of Research, at Colentina Hospital, Soseaua Stefan cel Mare 19-21, sector 2, 020125 Bucharest, Romania. (President of the Colentina Hospital Ethics Committee of Research: Gheorghe Andrei Dan, andrei.dan@gadan.ro), for researchers who meet the criteria for access to confidential data.

**Funding:** The authors received no specific funding for this work.

**Competing interests:** The authors have declared that no competing interests exist.

## Introduction

Modern life, with all the inherent progresses made in the health and science fields, made everyone think that pandemics were left somewhere in the past. Even though once in a few years we are facing epidemics like Severe Acute Respiratory Syndrome, novel Influenza A/H1N1, Middle East Respiratory Syndrome, which put people through a state of high anxiety, these were limited and didn't leave behind a high number of deaths. But at the end of 2019, a novel virus emerged in Wuhan, China, which put everything into a whole new perspective, since this virus spread all over the world and threatened the life of every human being on this planet. At this moment, there are 7.387.386 cases reported in the whole world and 415.778 deaths [1], while the SARS pandemic, the first pandemic of the 21st century, which emerged from Guangdong province in China in 2002 lead to 8098 cases and 774 deaths [2, 3]. Each of these outbreaks raised problems for the healthcare workers (HCWs), not only on a physical level, but also on a psychological one, due to their fear of contaminating themselves or their dear ones. Previous studies, which aimed to assess the impact of COVID-19 pandemic on HCWs, have shown that they experienced insomnia, moral injury, anxiety, distress and depression; negative emotions were present at first, and positive emotions appeared gradually [4, 5]. Sun N et al [5] evaluated in a qualitative study the psychological impact of COVID-19 on nurses, while Liu Q et al [6] assessed the way in which nurses and physicians experienced the early stages of the pandemic. Psychological outcomes were evaluated through various scales in surveys, by Lai J et al [4] on HCWs from China, by Tan BYQ et al [7] on HCWs from Singapore and by Chew et al [8] on HCWs from Singapore and India; all three of them showed that HCWs experienced anxiety and depression. Until now, no qualitative study included in their sample residents; we have to take into account the fact that the residents are young, with less experience and they may be more prone to experience a pandemic crisis in a negative way; it has been shown that younger age and being a junior is associated with a higher risk of psychological distress [9]. None of the studies focused on the preparedness period of COVID-19, and none of the studies was set in Europe. The preparedness period is represented by the phase when there is no direct contact with the threat, but all efforts are directed towards anticipating the course of events, preparing responses to those events, and trying to translate those responses into practice. We believe that this period, which is characterized by a state of expectation, is one of the hardest to endure. We aimed to explore the way in which the HCWs from two COVID-19 units from Bucharest, Romania experienced the preparedness period of the pandemic.

## Materials and methods

### Aim

The aim of this study was to evaluate, and to understand how HCWs experienced the preparedness period of COVID-19 pandemic in order to ascertain the perceived weaknesses and strengths regarding the way in which the Healthcare system, and the Healthcare units were organizing, and preparing for the pandemic response; consequently, those perceived weaknesses and strengths would reflect in the HCWs well-being. This is a mandatory process if we want to have a larger frame over the response to the pandemic, to gain a better understanding of the experience, and to improve future pandemic responses by preventing the perpetuation of the same errors (perceived weaknesses).

### Methodology

The research was qualitative exploratory using semi-structured in-depth interviews. The COREQ criteria (Consolidated criteria for reporting qualitative research) were used to report the study methodology [10].

**Table 1. Interview topic guide.**

| |
|---|
| 1. How did you feel when you found out that a novel virus emerged in China? |
| 2. How did you feel when you found out that there were confirmed cases in Romania? |
| 3. How did you feel when you found out that the hospital in which you work was going to be transformed in a COVID-19 unit? |
| 4. How do you think that the pandemic will impact the way in which you currently practice your work? |
| 5. What are your main concerns with regard to the current situation? |
| 6. What would you change if you could do this? |

Recruitment was directed via telephone or via e-mail; we sent to the participants the information form together with the consent form, and we let them decide if they wanted to participate. The interview consisted in 6 questions which are presented in "Table 1"; the questions evaluated how participants reacted when they found out about SARS-CoV-2, and how they perceived the experience. Additional questions were asked to ensure rich data collection. It was initially piloted on two persons in order to establish if the questions we had designed for the interview would provide the data we needed in order to investigate the way in which HCWs perceived the preparedness period; no modifications were made regarding the interview topic guide after we conducted the pilot study. The interviews were conducted face-to-face or over the phone, according to the participants' preferences, and lasted from 15 to 93 minutes. All the interviews were audio-recorded and transcribed verbatim by the interviewer, with the anonymization of the transcript (each participant was assigned with a code formed of three digits instead of using his/her name). 12 weeks after the article will be published, audio-recordings will be destroyed.

## Sample and data collection

Participants were not involved in the development of the research questions, study design and recruitment process; however, the research was designed to elicit their perceptions.

Participants were selected from two hospitals from Bucharest, which were turned into COVID-19 units. The interview was conducted before doctors had contact with patients with SARS-CoV-2 in order to assess the way in which they experienced the preparedness period of the pandemic. Both of the researchers work in the same ward, while one of them is the chief of the ward. Only two participants worked on the same ward as the researchers, but they expressed their desire to participate in the study once they found out about it.

Since the experience of the HCWs may vary according to the age, experience and the function occupied in the hospital, we purposely selected respondents from various positions: senior physicians, resident physicians and nurses.

Purposive sampling was directed towards achieving maximum variation in age and specialty. Infectious diseases and surgery specialists were excluded. Volunteers received no remuneration.

## Analysis

The research question was experiential and exploratory, so we conducted a primarily experiential form of thematic analysis, using an inductive, data-driven approach, while focusing on both latent and semantic levels. We followed the stages described by Braun and Clarke [11], which are: familiarization with all the data, generating initial codes, actively searching for the themes, reviewing potential themes, defining and naming themes and finally writing up the themes into a report. After familiarizing with the data, the interviewer generated the codes. A

training in interviewing, coding and thematic analysis was realized before starting the project. The interviewer generated the codes for the transcripts and presented them to the second author, who also was the supervisor of the study; together, we matched the codes into themes in seven meetings. The report was then written, and we sent it to five randomly selected participants in order to perform member checking. All of the selected participants endorsed the draft. We achieved data saturation after 11 interviews. We decided to include in our report codes not only based on the saturation principle, but also on the saliency analysis principle. Saliency analysis is described by Buetow in 2010 [12] and advocates the fact that codes of high importance are not the ones that recur systematically, but the ones that addresses real world problems, potentially important for the aims of the study.

## Statements

Both authors took care of patients with SARS-CoV-2. One of them contaminated himself with the virus and developed the disease. We acknowledge the fact that we may have been subjective while performing the analysis, especially when analyzing the data by the saliency principle. We applied this principle only once, when we included in our analyze the stringent need for maintaining ones team, which is compounded by people in whom one trust (this was reiterated only twice through the transcripts); we shared the same thoughts, and we were drawn towards this idea.

## Ethical considerations

Before starting the recruitment, the approval for the study was obtained from the Colentina Hospital Committee of Ethics, after they evaluated the participant information form and the consent form.

## Results

A total of 17 HCWs participated in our study. The age ranged from 27 to 70; further characteristics of the participants are presented in "Table 2".

**Table 2. Participants' characteristics.**

| Participants | | Numbers |
|---|---|---|
| Age | <30 | 6 |
| | 30–50 | 10 |
| | >50 | 1 |
| Gender | F | 16 |
| | M | 1 |
| Race | White | 17 |
| Function | Senior Physician | 8 |
| | Resident Physician | 6 |
| | Nurses | 3 |
| Specialty | Internal Medicine | 8 |
| | Cardiology | 1 |
| | Hematology | 3 |
| | Neurology | 3 |
| | Rheumatology | 1 |
| | Pneumology | 1 |

**Table 3. Overview of themes.**

| Themes title | Themes definitions | Subthemes |
|---|---|---|
| 1. Dealing with the unknown | This theme evaluates the multiple levels at which the healthcare workers perceived the lack of knowledge during their experience in the Covid-19 unit. Finding themselves in the position of not knowing what to expect and in a continuous state of anticipation was linked, for most of them, to a high state of psychological distress and anxiety. | 1.1. Ascertaining the unpreparedness |
| | | 1.2. Fear of losing control |
| | | |
| 2. Humans versus doctors | This theme explores the internal fight of the HCWs between their own needs and their patient needs. | 2.1. Duty to treat |
| | | 2.2. The universal fear |
| 3. Sense of helplessness | Lack of honest communication with their superiors and authorities, while they were depending on them for protection, made them perceive that they were helpless. Besides, the disappointment provoked by their colleagues further contributed to their emotional imbalance. | 3.1. Dismay governs the day |
| | | 3.2. Dependability on an ill healthcare system |
| | | |
| 4. A bridge to heaven | HCWs understood, more or less consciously, that their well-being is in their power and they focused their attention, at least partially, towards factors that helped them in the process of recovering their state of mind. | |
| | | |

After we analyzed and coded the transcripts, we identified four themes: Dealing with the unknown, Humans versus doctors, Sens of helplessness and A bridge to heaven; each theme is defined in "Table 3". Two subsequent subthemes, detailed below, were attributed to the first three themes. Even though our aim was to asses both weaknesses and strengths, our participants were overwhelmed by negative feelings, so they did not provide us enough information regarding the strengths perceived by them.

## 1. Dealing with the unknown

**1.1. Ascertaining the unpreparedness.** From the very first moments, when hearing that a novel coronavirus emerged from China, where it led to a high number of cases and deaths, and that the virus had spread to Europe (Italy), the HCWs divided into two categories according to their opposite reactions at the acute stressor agent. Most of them, unconsciously urged into denial, one of the main defense mechanisms when it comes to handling difficult events; they simply blocked the external events from awareness, refusing to ascertain or to analyze the gravity of the situation, in order to prevent the anxiety/apprehension to arise. For example, in order to avoid dealing with the ugly truth, and to avoid figuring a way to overcome the difficult period that was about to come, this nurse preferred to believe that the information about the new coronavirus were not true: "I thought that nothing is true and that everything is fake news [. . ..] I said nooooo, there is a lie" (Nurse 3)

On the contrary, others fully perceived the actual, and the future magnitude of the situation, and maybe they even hyper perceived it; they skipped any form of defense mechanisms, so that they rapidly dug into the aforementioned states of anxiety/apprehension, as quoted below:

"I was anxious and scared of everything that was following, not since we had the first cases in our country, but from January and after that in February, when the first patient in Italy appeared, when I started to have paroxysmal crisis of anxiety and fear" (Senior-Physician 8).

One of the contributors to these extreme attitudes, was the fact that they felt completely unprepared for the emergent crisis, or any crisis whatsoever. None of them had previous trainings regarding donning and doffing, and this lack of training was perceived at the origin of their lack of awareness towards this kind of events. As it is illustrated below, even though the disaster was approaching, they did not know how to properly use the equipment in order to

protect themselves and the others, while having these necessary steps imprinted in the routine is mandatory in emergency situations, in order to avoid wasting life-saving time, and building this into one's routine takes time.

> "I heard an epidemiologist from the army speaking, who said that every doctor should be trained to do this, and that is absurd for us to ask for this only now. This draw my attention on the fact that we were never trained, not during our studies in the university, not in real life, not as time went by [. . .]. Yes, we should have been trained periodically, because, look, such a situation could arise anytime, and in fact we are totally unprepared. [. . .] And it's in vain to look a thousand times at videos now, because until you don't do it, you can't fit it into your pattern of behavior." (Senior-Physician7)

HCWs felt overcome in this situation. Firstly, because their contamination, and subsequently the contamination of their dear ones were depending on developing high skills regarding proper donning and doffing in short time, which should have been developed in time, by regular trainings; secondly, as it is shown below, because the availability of the personal protective equipment (PPE) during the crisis was under a question mark, and even though they tried to overcome this lack by different practices, like buying PPE on their own, they knew that these necessary combat arms should have been assured by others, by that moment.

> "I spent on my own about 3000 euros with PPE, scrubs. . . I bought in desperation masks and alcohol for disinfection [. . .] You have to understand the fact that they transformed us into a COVID unit and nobody taught us how to put on and remove the PPE, we searched for videos and started to learn alone and supervise each other." (Senior-Physician 8)

**1.2. Fear of losing control.** The fear of the unknown was directly linked to the fear of losing control. Commitment to scientific knowledge is one of the top responsibilities in this profession; over the years, not only doctors, but all the HCWs got accustomed with knowing what they are dealing with, with knowing that they can always search for the information, look up in the guidelines, or ask a more experienced colleague in case of a dilemma. So, when doctors had to face this unprecedented situation, their acute responses linked to their fear of losing control, to the fear that there will no longer be available any entities to validate their actions and decisions like it was before.

> "I have this constant restlessness about the fact that we will not know what to do, that we do not have guidelines or protocols and that. . .we will have to juggle with the situations and, besides that, maybe at some point we will have to take care of intubated patients, and I have this big anxiety related to that because it is one thing to be an ICU specialist and another one to be a specialist which doesn't deal with this kind of patients". (Resident-Physician4).

As quoted above, doctors were put in the situation of having to walk in the dark of the unknown, of having to take full responsibility on "guessing" the best treatment for an unstudied disease, or how to correctly manage unprecedented situations. Even though in medicine there are many areas defined by uncertainty, when good professional judgement makes the difference, doctors experienced a high level of distress when they realized that they will have to assume the fact that their decisions are based on uncertainties, that they do not know what is in the best interest of their patients. When their sine qua non principle "do no harm" was torn into pieces and transformed into "you will do harm in order to have the chance to do good" their mental wellbeing was disrupted and a breach was created.

## 2. Humans versus doctors

**2.1. Duty to treat.** Most of the members of the healthcare staff had an internal conflict, and every day that passed by, they had to try to balance it; they had to balance between what it would be right to do, and what they would want to do, between who needs them the most on the short term, and on the long term, their patients or their family, knowing that if they choose their patients, they may never come home.

This was particularly acute for the HCWs who also had a moral obligation towards members of the family, like parents had for their children, or like sole children had for an old, sick parent.

"I obviously thought about resigning and being a little deserter, I took them all into account thinking at her safety (her daughter) not at my profession, my vocation. There were voices that said that if we resign now, we will have our right of free practice suspended. And I would like to be a physician after COVID-19 will go away. I can only hope, and do what is up to me in order to not contaminate myself and to find a way to go back to my family". (Senior-Physician 7)

For example, this doctor experienced the implications that being in the front line of a pandemic had over family members, in such an acute way, that he/she even questioned his/her meaning in life, and his/her ability to practice this profession. No matter the decision, his/her mental well-being would consequently be impacted in a negative way, and once the decision to accept, and to fulfill professional duties in the pandemic was made, balancing between self-protection and the protection of his/her dear ones on one side, and not neglecting the patients on the other side, became a daily burden chore.

**2.2. The universal fear.** The main fear of every HCW was represented by the universal fear of the humanity, the fear of death. By the nature of their profession, HCWs are struggling with death more often than others, and they usually manage to overcome this negative side by "hiding" under the armor of professionalism and objectivity. But their previous experiences made not much of a difference when they had to face the imminence of their death, or of the death of their beloved ones. Nothing prepares someone for this kind of situations.

"Oh, no. I am not afraid of death. I had a form of neoplasia and if that didn't kill me, this thing is neither going to kill me. And if this kills me, then it kills me and this is it. But it would be much worse to know that my children will die or that my husband will die. It is. . . awful." (Nurse 1)

As it is shown above, it appears that for HCWs, accepting their own death was easier; they came at peace with the worst thing that could ever happen with their own person, and saw it like a form of liberation. It was much harder, though, for them to reconcile with the thought that their dear ones may die; this may be due to the fact that, as HCWs, they may had felt more responsible for the fate of their beloved ones. HCWs are saving lives on a daily basis, and the thought of failing to do this for their dear ones was simply unbearable. More than for anyone, and more than anytime, they should have been in control over the evolution of the disease, and once again they had lost control.

## 3. Sense of helplessness

**3.1. Dismay governs the day.** When their work implied exposing themselves to serious risks of harm and even death, one of the most important things for them was to feel that the

structures involved in assuring their well-being were there, taking care of them while performing their job; this could have been ascertained only by direct communication with them, and by observing not only words, but also effects. They needed an honest communication in which the authorities would have spoken freely about the severity of the situation, about the flaws in the system, and about what they should expect for in the future; only such a reaction could help break the unknown and comfort them, because most often living with the unknown is worse than the gravity of the situation.

> "I have this fear that we are not informed. They are hiding things from us. At first, they told us that we are reporting everything to the World Health Organization, that they are surveilling us, that we report everything, but I have the feeling that we do not report everything. It's only what I think, but I have this feeling that they would not transform more and more hospitals into COVID units, if they didn't have information that the situation is actually much more serious. And the lack of communication was from my direct superiors. Yes, what could make me feel safer is a better communication with my superiors which theoretically should know better." (Senior-Physician 7)

Instead, as it is quoted above, the HCWs were disappointed by the lack of communication between them and their superiors, and between their superiors and the authorities. Their constant assurances that everything is going according to the plan, while nobody knew what the plan actually was, worsened their state of mind during the preparedness period, it made them think that all these guarantees were due to the fact that the authorities were trying to hide from them a much worse situation.

On top of this, the state of unresolvable unknown which governed the situation, was exacerbated not only by the way in which the authorities managed the situation, but also by their alleged team, as it is shown below. They felt disappointed by their colleagues, which refused to work in a COVID-19 unit, giving them a sense of lack of support; they felt abandoned by the ones that should be in their team, making them to highly perceive that they can count on nobody.

> "There were some colleagues which refused to work in a COVID unit, people who knew someone and most of the doctors from the Institute, which were not elderly nor with comorbidities, they were as healthy as someone can be, but from various reasons like the fact that they are sole parents or I don't know what else, they refused [. . .] from 130 doctors which work in our hospital, only 35 were assigned to work in the COVID unit." (Senior-Physician 8)

**3.2. Dependability on an ill healthcare system.** When the literature presented as a potential treatment, drugs that there were not at their disposal, they felt that no matter how much they will read, and no matter how much they will struggle, all their work would have been in vain. The patient-physician interaction would no longer imply saving lives, and HCWs felt like they would be transformed into observers of the hazard while performing a tainted medical act.

> "Well, there are some drugs that are completely unavailable, like remdesivir. You have to give tocilizumab if interleukin-6 is elevated but we do not perform this test. I mean, there are things in the protocols that you can't do, let's say that the protocol is not final, that it changes, but in the protocol, you have things that you can't do." (Senior-Physician 3)

Besides, doctors had to deal with the responsibility towards their regular patients, because when hospitals were transformed into COVID units, the authorities didn't provide support for these patients, chronic patients which needed monthly prescriptions, which were prescribed only in hospitals.

> "What was really hard, what had a big impact, was seeing that all your patients which were counting on you and which were calling you, couldn't come to you anymore and you had to say to them that they have to go somewhere else. It's complicated, because you follow them from a long time, you have a certain therapeutic protocol, there is a sort of confidence that has formed between you two, and suddenly you have to say to them something like <<I can see you only if you get infected with the coronavirus>>." (Senior-Physician 3)

As quoted above, when they had to abandon their patients in difficult times, they felt that they had violated their ethical code, and this put the HCWs to a state of psychological imbalance, especially because those were unwanted actions that they had to make, which could have been avoided if the authorities would have been more involved. Now, they had to deal not only with the fact that they could not actually treat COVID-19 patients, because there was no treatment, but also with the fact that they could no longer treat the ones that were actually treatable. Once again, the healthcare system let them down when they needed it.

## 4. A bridge to heaven

The preparedness period was characterized by an emotional rollercoaster which was difficult to ride, and HCWs had to constantly seek for things which might comfort them. In this time of changes, in order to feel some form of alleviation, they had an urge to maintain something from the way they previously performed their job, something that could connect the past and the present, a "bridge" between the period when they were in a state of mental well-being, and the actual moment when they were in an emotional distress.

> "What I fought for in my ward was to maintain the old structure, to remain as close as we could to all the things that we did before, because is very hard to get rid of all your habits, this completely turns you upside down." (Senior-Physician 3)

> "What could make me feel safe would be to have with me my team, that I can count on." (Senior-Physician 7)

This "bridge" was represented by different entities, like changing as few things as possible while creating routes for transporting the patients, or working on shifts with the doctors whom they knew and used to work with, as was presented in the previous quotes. Routine was desirable not only in regard with donning and doffing, but also in a broader way, because they felt that maintaining at least a piece of their routine, brought them one step closer to the optimum conditions in which they could properly perform their work, to an environment which provided them an equilibrium between the ongoing disaster and a certain state of peace.

Besides struggling with accelerated changes regarding their everyday practice and environment, they also had to deal with an unfamiliar disease, and this is an exceptional task which HCWs have to perform while crossing pandemic times. In the era of evidence-based medicine, doctors are used to manage and treat patients following information from the guidelines, from randomized controlled trials, or at least from small, observational studies. Now, in the beginning of the pandemic, when almost nothing was known, they could no longer care for patients

under the stigma of high-quality data. At that moment, fear was all over around, and fear was aggravated by the lack of information.

> "I started to read about viruses generally speaking and. . .after all, historically speaking, there had been before pandemics which have drastically reduced the population of the earth." (Resident-Physician 6)

> "I read everything that was published. Every morning after I woke up, I stayed in bed for another hour to read all the literature that has been published." (Senior-Physician 8)

So, as it is shown above, in the beginning, in the absence of data, they started to look for facts from the past; they knew that any form of knowledge in these times would make them feel that they regained, at least partially, the control over the situation, and this alleviated them. As time went by, they found a support pole in any piece of trusted information, a form of future validation, not only in information regarding diagnosis and treatment, but also in general information like organization of the wards and teams. Knowledge is panacea, and knowledge, along with maintaining a form of routine, represented a bridge between new times and old times, between anxiety and hope, between a disaster and the satisfaction of overcoming a pandemic.

## Discussions

COVID-19 pandemic represented, represents and will still represent a challenge for every country and for the HCWs from all over the world. Besides, pandemics with novel infectious agents can emerge in the future. Even though over the last decades we have experienced several outbreaks and we would have expected to a certain degree of awareness by now, while having efficient training programs implemented, it appears that this is not the case. None of the participants in our study had received appropriate trainings in regard to donning and doffing, not before the pandemic state and neither after that. Entering in an isolation ward while not having previous experience with infectious diseases is perceived as an important stressor [6]; having to do that without being trained or while being insufficiently trained is even more stressful. The lack of training led to a perceived state of unpreparedness. It is well known that preparing the staff for associated job challenges reduces the risk of mental distress [13]; proper trainings should be implemented in every hospital in the expectative of a crisis situation in order to maintain HCWs awareness. On top of this, all the HCWs experienced a constant state of fear, fear of a possible contamination, fear of death, fear for their beloved ones or fear of a future lack of PPE. The latter was due mainly to an inefficient communication with the superiors and the authorities. One of the HCWs requests during the COVID-19 pandemic was "to be heard" [14]; they wanted to have the means necessary to address opinions to higher commissions, opinions which should be analyzed and taken into account. Authorities should engage in honest and meaningful discussions, without false reassurances [15, 16]. Besides, HCWs felt the need to be protected and supported. Psychological aid during this period was considered mandatory and different institutions strived to provide this kind of support for HCWs. But, according to Chen et al [17], even though nurses presented signs of irritability and psychological distress, they stated that they do not have any problem, and consequently refused any form of psychological support. Therefore, is it enough to provide psychological assistance during difficult times, or maybe this is not enough due to the fact that HCWs do not admit that they need help, or because they are reluctant to express their worries? Maybe this points out that a climate of psychological safety among HCWs should have been consolidated, before pandemic times, so that they can feel comfortable with speaking up with questions, concerns, ideas or mistakes of them or of their superiors.

The virtue of altruism marks our profession, but self-preservation and preservation of our dear ones represents instinctive behaviors of all humans, especially when unexpected adverse events threaten to arise. The primacy of patient welfare embodies the whole foundation of the clinical medicine. But are those principles still standing up in pandemic times? Do doctors have a duty to treat? Our study points out the fact that there was a conflict between the HCWs which decided to step-out from the fight, especially when their absence wasn't due to justified reasons, and the HCWs which remained in the fight. Bensimon et al reported in their study that some people consider that HCWs have an unlimited duty to care and to treat [18]. But, as Ashcroft recently alluded, it is reasonable to keep older HCWs off the COVID-19 frontline [19]. Besides, HCWs with comorbidities should also be spared. But what is the age threshold and which comorbidities are justifying this? Who decides this? Iniquitous decisions may result in moral injury for representants from both sides of the line and even though these issues have previously been reported, it appears to having remained unsolved. In consistence with previous studies [5, 20], HCWs experienced fear and anxiety while thinking at their beloved ones; they found easier to accept their own death than the death of a family member.

One of the main concerns of the HCWs was with regard to the way they will manage to provide to their patients a proper patient-physician interaction, while maintaining their own health. It is crucial to explore the balance between the clinicians' personal safety, and their patients' needs during pandemic times. Appropriate standards of care should be further stated by the authorities [21–23]; HCWs should be supported and their decisions should be validated. As far as we know, this is the first study to assess the way in which COVID-19 pandemic preparedness period was experienced by HCWs from an European country. Our sample of participants included not only physicians and nurses, but also residents, while it is known that age influences the way in which a crisis situation is experienced, younger age and lack of experience being linked with a higher risk of developing negative emotions and psychological imbalance [9]. Evaluating the factors that led to psychological imbalance during the preparedness period of COVID-19 pandemic enables the process of improvement in regard to the preparedness period of a future crisis. Conducting qualitative research with HCWs regarding their perceptions and concerns during pandemic times may help in informing larger policy decisions; this may lead, for example, to the implementation of periodic trainings for emergency situations, which may confer a certain degree of preparedness to HCWs, that will permit to face more confident future pandemics.

## Limitations

The main limitation of this study consists in the fact that it was developed at a single moment in time; therefore, a longitudinal qualitative study, in order to analyze the evolution of their emotions, from the preparedness period, to the active phase of the pandemic, and afterwards to the end of the first wave would be necessary to evaluate if HCWs consider that their fears from the preparedness period were justified, and also to inform what were the gaps and weaknesses in the actual practice, in order to raise awareness and to facilitate an address to them. Another limitation resides in the fact that our participants sample was represented in majority by female subjects, while it is known that emotional stress may be experienced in different ways, according to gender [24]. The fact that we interviewed only people from two hospitals may also represent a limitation, given the fact that trainings regarding crisis situations may differ from institution to institution.

## Acknowledgments

We thank all healthcare workers who participated in our study.

## Author Contributions

**Conceptualization:** Cristian Baicus.

**Data curation:** Laura Elena Stoichitoiu.

**Formal analysis:** Laura Elena Stoichitoiu, Cristian Baicus.

**Investigation:** Laura Elena Stoichitoiu.

**Methodology:** Laura Elena Stoichitoiu, Cristian Baicus.

**Project administration:** Cristian Baicus.

**Resources:** Cristian Baicus.

**Supervision:** Cristian Baicus.

**Writing – original draft:** Laura Elena Stoichitoiu.

**Writing – review & editing:** Cristian Baicus.

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
