## [Decision Letter · Decision Letter 0]

19 Aug 2020

PONE-D-20-18668

COVID-19 pandemic preparedness period through healthcare workers’ eyes: a qualitative study

PLOS ONE

Dear Dr.Stoichitoiu,

Thank you for submitting your manuscript to PLOS ONE. After careful consideration, we feel that it has merit but does not fully meet PLOS ONE’s publication criteria as it currently stands. Therefore, we invite you to submit a revised version of the manuscript that addresses the points raised during the review process.

Please address the issues raised concerning methodology, results and discussion raised by the reviewers.

We look forward to receiving your revised manuscript.

Kind regards,

Rosemary Frey

Academic Editor

PLOS ONE

Journal Requirements:

2. The title of the submission should specify that the research is representative of a Romanian healthcare facility.

3.We note that you have indicated that data from this study are available upon request. PLOS only allows data to be available upon request if there are legal or ethical restrictions on sharing data publicly. For information on unacceptable data access restrictions, please see http://journals.plos.org/plosone/s/data-availability#loc-unacceptable-data-access-restrictions.

Reviewers' comments:

Reviewer's Responses to Questions

**Comments to the Author**

1. Is the manuscript technically sound, and do the data support the conclusions?

Reviewer #1: Yes

Reviewer #2: No

2. Has the statistical analysis been performed appropriately and rigorously? 

Reviewer #1: N/A

Reviewer #2: No

3. Have the authors made all data underlying the findings in their manuscript fully available?

Reviewer #1: No

Reviewer #2: No

4. Is the manuscript presented in an intelligible fashion and written in standard English?

Reviewer #1: Yes

Reviewer #2: Yes

5. Review Comments to the Author

Reviewer #1: Thank you for the opportunity to review your manuscript. The topic is important and will continue to be so for a long time. It would have been useful to explain for international readers what you mean by 'the preparedness period of COVID-19.

Your research question was clearly stated. However, whilst your aim was to consider perceived weaknesses and strengths, the latter was missing from your analysis and discussion. Your appraoch to thematic analysis was clear and use of Table 3 illustrated the themes and subthemes. Quotes were used to illustrate the themes. You stated that saturation of data occured after 11 interviews (line 112) and providing some further explannation of this would be useful. In addition, if saturation occured after 11 interviews what was the purpose of conducting six more interviews? Did your analysis of the residents interviews highlight more negative experiences as you suggest this might be the case in your introduction.

I note that on line 314 you refer to bullying, and this is quite a strong statement, but no quotes are provided to support this. In your conclusion (line 417) you state that further attention should be paid to factors that may help HCWs regain their balance, but what does that mean?

I did wonder whether some discussion of 'uncertainty' and also 'psychological safety' could have added more indepth understanding to your discussion.

An additional paper that might be helpful to you is the following;

Shanafelt, Et. et al (2020) Understanding and addressing sources of anxiety among health care professionals during the COVID-19 pandemic. Journal of the American Medical Association April7th.

Reviewer #2: This is a narrative or a commentary rather than a piece of scientific research. The authors have presented a series of responses to a standard set of interview questions in a journalistic format with no solid conclusion drawn. No information was provided about the number of eligible subjects and the response rate. Participants were voluntary and therefore selection bias is inevitable. With a sample size so tiny and such skewed demographics, it is impossible to draw any conclusions whatsoever.

6. PLOS authors have the option to publish the peer review history of their article (what does this mean?). If published, this will include your full peer review and any attached files.

Reviewer #1: **Yes: **Susan Waterworth

Reviewer #2: No

---

## [Author Response · Author response to Decision Letter 0]

1 Sep 2020

Reviewer #1: Thank you for the opportunity to review your manuscript. The topic is important and will continue to be so for a long time. 

It would have been useful to explain for international readers what you mean by 'the preparedness period of COVID-19.

We added the following explanation in the Introduction section “. The preparedness period is represented by the phase when there is no direct contact with the threat, but all efforts are directed towards anticipating the course of events, preparing responses to those events and trying to translate those responses into practice”. 

Your research question was clearly stated. However, whilst your aim was to consider perceived weaknesses and strengths, the latter was missing from your analysis and discussion.

Yes, this was our aim. But participants were overwhelmed by negative feelings, so they did not provide us enough information regarding the strengths perceived by them, at least not enough information in order to generate a code or a theme. We added this in the results section.

Your approach to thematic analysis was clear and use of Table 3 illustrated the themes and subthemes. Quotes were used to illustrate the themes. You stated that saturation of data occured after 11 interviews (line 112) and providing some further explannation of this would be useful. In addition, if saturation occured after 11 interviews what was the purpose of conducting six more interviews? 

We performed more than one interview per day and we did not manage to transcribe every interview immediately after recording it. Therefore, we had already taken the last six interviews when we realized that we had already reached data saturation. Therefore, as a gratitude gesture for the time offered by our colleagues, and their involvement in our project, we decided to include them in the study. 

Did your analysis of the residents interviews highlight more negative experiences as you suggest this might be the case in your introduction.

Yes, there were some differences regarding residents’ emotions and experiences, when compared with those of the seniors. We are aware that including residents in our study may have influenced the codes and the themes, but our aim was not to analyze their experiences by comparison with seniors’ experiences, but rather to ascertain an overview of the HCWs emotions and experiences.

I note that on line 314 you refer to bullying, and this is quite a strong statement, but no quotes are provided to support this.

Bullying was not present expressis verbis in our transcripts; it transpired indirectly during the interviews. We also reanalyzed the transcripts; reading them months after writing them, and after dealing with Covid-19, we realized that this was, as you said, a strong statement, which might have been influenced by our emotional status at the moment. Therefore, we decided to delete that part from the results section. 

In your conclusion (line 417) you state that further attention should be paid to factors that may help HCWs regain their balance, but what does that mean?

We rephrased that paragraph as it follows: “Identifying sources of support during this period, by asking HCWs what they needed , what helped them or what would have helped them, and not by searching for a feedback in regard with alleged forms of reinforcement, should be further assessed in longitudinal qualitative studies.”

I did wonder whether some discussion of 'uncertainty' and also 'psychological safety' could have added more in depth understanding to your discussion.

An additional paper that might be helpful to you is the following;

Shanafelt, Et. et al (2020) Understanding and addressing sources of anxiety among health care professionals during the COVID-19 pandemic. Journal of the American Medical Association April7th.

We thank you for your recommendation; we read the paper, and we further detailed the discussion section. 

Reviewer #2: This is a narrative or a commentary rather than a piece of scientific research. The authors have presented a series of responses to a standard set of interview questions in a journalistic format with no solid conclusion drawn. No information was provided about the number of eligible subjects and the response rate. Participants were voluntary and therefore selection bias is inevitable. With a sample size so tiny and such skewed demographics, it is impossible to draw any conclusions whatsoever.

In qualitative research, in order to identify themes and express results, the authors immerse into data; therefore, we consider that a narrative or a commentary component, as you have stated, which is also subjective, is inherent to any qualitative study, by its own nature. Neither COREQ nor SRQR guidelines for reporting qualitative research stipulate to provide information about the number of eligible subjects and the response rate; as far as we know, these requests are necessary when reporting results of surveys, in quantitative research. Participants had to be voluntary, because nobody can constrain someone to enroll in a study, nor to offer an interview. We understand that it may have been a selection bias, but we believe that a certain degree of selection bias is present in every qualitative study. Regarding the sample size, some scientific papers say that saturation appears after twelve interviews. We cannot say that this is universally true, but is certain that qualitative research tends to use smaller samples size than quantitative research. Besides, according to the scientific literature, sample size is in close connection to the concept of saturation, and remains a widely used rationale for sample size in qualitative research; therefore, when saturation occurs, the sample size is enough, no matter the number of participants enrolled in the study. In our case, saturation occurred after 11 interviews, therefore 11 participants would have been enough. In regard with demographics, qualitative researchers may aim to sample for diversity of perspectives, or for typicality or homogeneity of perspectives; as far as we know, sampling for heterogeneity (in our case, age heterogeneity), is not a mistake, nor a weak point in a qualitative study. Our aim was not to identify patterns through various age categories, nor to compare them, but to have an overview of the health care workers emotions and experiences through this difficult time.

---

## [Decision Letter · Decision Letter 1]

13 Oct 2020

PONE-D-20-18668R1

COVID-19 pandemic preparedness period through healthcare workers’ eyes: a qualitative study from a Romanian healthcare facility

PLOS ONE

Dear Dr.Stoichitoiu,

Thank you for submitting your manuscript to PLOS ONE. After careful consideration, we feel that it has merit but does not fully meet PLOS ONE’s publication criteria as it currently stands. Therefore, we invite you to submit a revised version of the manuscript that addresses the points raised during the review process.

Please address Reviewer 3 comments regarding methods and findings.

We look forward to receiving your revised manuscript.

Kind regards,

Rosemary Frey

Academic Editor

PLOS ONE

Reviewers' comments:

Reviewer's Responses to Questions

**Comments to the Author**

1. If the authors have adequately addressed your comments raised in a previous round of review and you feel that this manuscript is now acceptable for publication, you may indicate that here to bypass the “Comments to the Author” section, enter your conflict of interest statement in the “Confidential to Editor” section, and submit your "Accept" recommendation.

Reviewer #1: All comments have been addressed

Reviewer #3: (No Response)

2. Is the manuscript technically sound, and do the data support the conclusions?

Reviewer #1: Yes

Reviewer #3: Partly

3. Has the statistical analysis been performed appropriately and rigorously? 

Reviewer #1: N/A

Reviewer #3: N/A

4. Have the authors made all data underlying the findings in their manuscript fully available?

Reviewer #1: No

Reviewer #3: Yes

5. Is the manuscript presented in an intelligible fashion and written in standard English?

Reviewer #1: Yes

Reviewer #3: Yes

6. Review Comments to the Author

Reviewer #1: (No Response)

Reviewer #3: First, I want to comment on Reviewer 2's critiques, which I found to be laughable. Thematic analysis is science and is not 'journalism.' It doesn't take a genius to know there are methods beyond run of the mill positivism. Google it.

More specific to this paper, I do believe the study has potential, but there are some issues that need to be addressed prior to publication. I point these out by line number below.

62-why is the 'preparedness period; the hardest to endure? I would think it would be the period in which the disaster - that being the actual instance of pandemic would be the most difficult. Please clarify.

70-'weakness and strengths' of what specifically?

71-please describe why it is a 'mandatory process' and what 'errors' you are talking about - are they those mentioned in the section above? If so, please state explicitly so the reader doesn't get off track or confused like I did. Also, why was doing this in Europe, specifically in Romania, important?

73-COREQ Criteria - What is that? Please define and describe for unfamiliar readers. Also, why did you choose a qual study? I understand you mention there were no prior qual studies, but this in itself doesn't mean there should be. The need for such research is driven by the current state of the work and gaps left in them - what is the specific knowledge gap this fills? Why do your question(s) necessitate a qual approach? The method is driven by the questions and the need to answer them/it. So, tell the reader why this work was crucial.

79 - What do you mean by 'rich' data? Why is it important?

80- What sort of validity are you trying to establish because it is different given your methods. Maybe you are talking about trustworthiness? It might be helpful to refer to some seminal texts here (I know some specific methods texts are talked about later). Glaser, Strauss, and Corbin and their variety of works might be useful in this section and will help fend off the qual hater trolls that always come abut try to wreak shenanigans on these studies.

107-What do you mean by "was realized?"

My next comments center more on the themes and findings of the work. I am going to point towards lines 167-169 as to what I see as the most interesting contribution in what is a somewhat messy bunch of data which indicates the analysis isn't quite yet done. The point raised in those lines (and of course, I have not seen or analysed the actual data) indicates that a breeching experiment (a la Garfinkle) has indeed been performed to some extend via the pandemic. What this accomplished is an upending of reality and the illusion of control that not only medicine has over pathogens, but also humans have over death (and by extension medicine has over death). So this is much more than 'shades of grey' - which you might reconsider the title given that wretched book and movie - COVID has torn apart those illusions of control and people, including the HCW you studied and are a member of, have been thrust into. You might want to read the following paper: https://www.academia.edu/20128995/Disaster_Preparedness_as_Social_Control - it is a different case but the methods and themes are related.

I think this larger frame might help make sense of all this stuff you have going on here, which as it stands is a lot to digest. I do believe more analysis is necessary even if saturation was reached (totally possible at your # of interviews, no question), themes should be condensed, and the way you describe your data within themes needs more contextualization - you just kind of throw quotes there and don't really massage them into findings and conclusions for the reader - you must do this for it to make sense. If so, you will have a really great, publishable paper. I know it is somewhat discouraging to go back to analysis, but I speak from experience when I say, I have done this numerous times and it has only made my work that much better.

7. PLOS authors have the option to publish the peer review history of their article (what does this mean?). If published, this will include your full peer review and any attached files.

Reviewer #1: **Yes: **Susan Waterworth

Reviewer #3: **Yes: **Natalie D Baker

---

## [Author Response · Author response to Decision Letter 1]

29 Oct 2020

First of all, we want to thank the third reviewer, for her important commentaries, and her guidance.

1. First, I want to comment on Reviewer 2's critiques, which I found to be laughable. Thematic analysis is science and is not 'journalism.' It doesn't take a genius to know there are methods beyond run of the mill positivism. Google it.

More specific to this paper, I do believe the study has potential, but there are some issues that need to be addressed prior to publication. I point these out by line number below.

2. 62-why is the 'preparedness period; the hardest to endure? I would think it would be the period in which the disaster - that being the actual instance of pandemic would be the most difficult. Please clarify.

We believe that the preparedness period is the hardest to endure, much harder than the actual pandemic phase, because during preparedness you are, at best, constantly exposed to all sorts of news and information from the experience of others, and since you are not involved “in the fight”, you are only left with analyzing and overthinking all those information, which is not always productive, but rather it represents a huge source of anxiety. At worst, you know nothing about the disease, the rate of transmission, the mortality, the actual effectiveness of the personal protective equipment, and one can only estimate all those things; during the time you wait for an emerging crisis to reach, some of us, if not most of us, tend to overestimate all those “magnitude parameters”, which is also a big source of anxiety. Our experience, and also our colleagues experience, confirmed our feelings, and once we were involved in the process of taking care of COVID-19 patients, we focused on doing our best on helping these people, rather than thinking at everything that was happening or would happen, and so the anxiety from the preparedness period diminished. 

3. 70-'weakness and strengths' of what specifically?

We aimed to ascertain the perceived weaknesses and strengths regarding the way in which the Healthcare system and the Healthcare units were organizing and preparing for the pandemic response; consequently, those perceived weaknesses and strengths would reflect in the healthcare workers well-being. We added this clarification in the Materials and Methods section. After our first revision, in accordance with the first reviewer commentaries, we added in the Results section the fact that the participants were overwhelmed by negative feelings, so they did not provide us enough information regarding the strengths perceived by them, at least not enough information in order to generate a code or a theme. 

4. 71-please describe why it is a 'mandatory process' and what 'errors' you are talking about - are they those mentioned in the section above? If so, please state explicitly so the reader doesn't get off track or confused like I did. Also, why was doing this in Europe, specifically in Romania, important?

Yes, the errors we were referring to, are represented by the weaknesses that they perceived; we added this in the paragraph, in order to clarify what we meant. We believe that this is a mandatory process, because only by analyzing the experience of the healthcare workers during pandemic time, we can understand what issues need to be addressed in the future, in order to improve the healthcare system response to a pandemic. Even though we have dealt with previous outbreaks, none of them had this magnitude, and clearly anticipating and planning a pandemic response in “neutral” times is not enough, which is reflected in the lack of personal protective equipment supplies, the lack of trainings regarding donning and doffing, and so on. Moreover, there are major differences between countries, and continents, regarding culture, habits, way of socializing, and clearly regarding the quality of the healthcare system. We believe that it is important to ascertain perceived weaknesses and strengths from various people and countries, in order to have a proper larger frame over the response to a pandemic, which will permit us a better understanding of the experience, and consequently, as we hope, it will lead to implementing some measures and improving future pandemic responses. 

5. 73-COREQ Criteria - What is that? Please define and describe for unfamiliar readers. Also, why did you choose a qual study? I understand you mention there were no prior qual studies, but this in itself doesn't mean there should be. The need for such research is driven by the current state of the work and gaps left in them - what is the specific knowledge gap this fills? Why do your question(s) necessitate a qual approach? The method is driven by the questions and the need to answer them/it. So, tell the reader why this work was crucial. 

COREQ criteria is the abbreviation for “Consolidated criteria for reporting qualitative research”, which represents one of the two reporting guidelines regarding qualitative research, suggested by the EQUATOR (Enhancing the QUAlity and Transparency of Health Research) Network, in order to improve the reliability and value of published health research literature; we added what COREQ stands for in the Materials and Methods section.

As we stated in the introduction, at the moment we performed, and wrote our study, there were two qualitative studies developed in China, one which enrolled nurses, while the other one involved nurses and doctors. As we mentioned above, there are major differences between China and other countries from all over the world, starting with the politic regime, and finishing with the healthcare system, and culture, which have a major impact in regard to how people respond to restrictive measures, dissemination of information, pandemic response and how people perceive all those facts. Taking all those mentioned above (and also those mentioned in our answer to the previous question), we believe that it is important to ascertain more points of view, in order to have a larger frame over how healthcare workers perceived the pandemic preparedness period (and for future studies how they perceived the pandemic response), in order to improve, on a global level, future responses. Besides, as we stated in the discussion section, our study is the first one to include residents, besides doctors and nurses, which, even though is inherent to a more heterogeneous information, and for some people, this may represent a minus, we consider that it is necessary in order to have a full view over the perceived weaknesses, moreover because residents were an important part in the response to the pandemic, at least in our country. We consider that a survey could not provide an in-depth understanding of the healthcare workers perceptions and feelings, maybe not even one with an endless number of questions (which would, of course, be unrealistic); we believe that qualitative research is the proper method to use when it comes to in-depth understanding and analyzing people emotions and feelings regarding sensitive issues. 

6. 79 - What do you mean by 'rich' data? Why is it important?

By rich data we mean detailed interviews, which offer information that permit in-depth understanding of the experience and proper analysis. Otherwise, there is not much difference between a qualitative study and a survey performed online.

7. 80- What sort of validity are you trying to establish because it is different given your methods. Maybe you are talking about trustworthiness? It might be helpful to refer to some seminal texts here (I know some specific methods texts are talked about later). Glaser, Strauss, and Corbin and their variety of works might be useful in this section and will help fend off the qual hater trolls that always come abut try to wreak shenanigans on these studies.

By validity, we meant that we wanted to evaluate if the questions included in the interview would be able to generate the data necessary for investigating how HCWs perceived the preparedness period. We rephrased that sentence in order to be clearer. We ensured trustworthiness by performing member checking. 

8. 107-What do you mean by "was realized?"

We meant that the person which conducted the interviews was prepared for this task by performing an extensive documentation. 

My next comments center more on the themes and findings of the work. I am going to point towards lines 167-169 as to what I see as the most interesting contribution in what is a somewhat messy bunch of data which indicates the analysis isn't quite yet done. The point raised in those lines (and of course, I have not seen or analysed the actual data) indicates that a breeching experiment (a la Garfinkle) has indeed been performed to some extend via the pandemic. What this accomplished is an upending of reality and the illusion of control that not only medicine has over pathogens, but also humans have over death (and by extension medicine has over death). So this is much more than 'shades of grey' - which you might reconsider the title given that wretched book and movie - COVID has torn apart those illusions of control and people, including the HCW you studied and are a member of, have been thrust into.

We are not quite sure if you understood that we used in our interviews Garfinkle’s research method, known as “breaching experiment”; if this is the case, we are sorry to inform you that we did not do this. We renamed the first theme, in order to avoid any form of association with that book, even an unwanted one.

You might want to read the following paper: https://www.academia.edu/20128995/Disaster_Preparedness_as_Social_Control - it is a different case but the methods and themes are related.

I think this larger frame might help make sense of all this stuff you have going on here, which as it stands is a lot to digest. I do believe more analysis is necessary even if saturation was reached (totally possible at your # of interviews, no question), themes should be condensed, and the way you describe your data within themes needs more contextualization - you just kind of throw quotes there and don't really massage them into findings and conclusions for the reader - you must do this for it to make sense. If so, you will have a really great, publishable paper. I know it is somewhat discouraging to go back to analysis, but I speak from experience when I say, I have done this numerous times and it has only made my work that much better.

We read with great interest the paper indicated by you, and we made some major changes in the Results section.

---

## [Decision Letter · Decision Letter 2]

22 Jun 2021

PONE-D-20-18668R2

COVID-19 pandemic preparedness period through healthcare workers’ eyes: a qualitative study from a Romanian healthcare facility

PLOS ONE

Dear Dr.Stoichitoiu ,

Thank you for submitting your manuscript to PLOS ONE. After careful consideration, we feel that it has merit but does not fully meet PLOS ONE’s publication criteria as it currently stands. Therefore, we invite you to submit a revised version of the manuscript that addresses the points raised during the review process.

Please provide the additional detail and updated references required by the reviewers.

We look forward to receiving your revised manuscript.

Kind regards,

Rosemary Frey

Academic Editor

PLOS ONE

Journal Requirements:

Reviewers' comments:

Reviewer's Responses to Questions

**Comments to the Author**

1. If the authors have adequately addressed your comments raised in a previous round of review and you feel that this manuscript is now acceptable for publication, you may indicate that here to bypass the “Comments to the Author” section, enter your conflict of interest statement in the “Confidential to Editor” section, and submit your "Accept" recommendation.

Reviewer #3: (No Response)

Reviewer #4: (No Response)

2. Is the manuscript technically sound, and do the data support the conclusions?

Reviewer #3: Partly

Reviewer #4: Partly

3. Has the statistical analysis been performed appropriately and rigorously? 

Reviewer #3: N/A

Reviewer #4: N/A

4. Have the authors made all data underlying the findings in their manuscript fully available?

Reviewer #3: Yes

Reviewer #4: No

5. Is the manuscript presented in an intelligible fashion and written in standard English?

Reviewer #3: (No Response)

Reviewer #4: Yes

6. Review Comments to the Author

Reviewer #3: 1. To clarify, I hope the authors understand this was a comment directed towards the prior reviewer and not the authors themselves.

2. I understand the authors points in response to my question and they are believable, especially since the authors define it as such:

"The preparedness period is represented by the phase when there is no direct contact with the

threat, but all efforts are directed towards anticipating the course of events, preparing responses to those events and trying to translate those responses into practice. We believe that this period, which is

84 characterized by a state of expectation, is one of the hardest to endure."

I am left with a couple important questions - 1) are the authors drawing on their own experiences too as they describe in their response letter? If so, then I suspect they have also engaged in a co-autoethnographic project, which adds even more legitimacy to this work (although is an often marginalized method). 2) how do they distinguish between the 'preparedness period' and the 'response period'? Perhaps, as authorities themselves of this context, these lines are blurry. Which in itself is extremely important as a finding and it worthy of mention. I am not trying to push more literature on to the authors or more work. However, I think 2 other papers that can help strengthen both the methods and findings, could be considered:

https://www.academia.edu/41122099/Hurricane_Harvey_Unstrapped_Experiencing_adaptive_tensions_on_the_edge_of_chaos

https://www.academia.edu/38022452/For_a_short_time_we_were_the_best_versions_of_ourselves_Hurricane_Harvey_and_the_Ideal_of_Community

The above works address the co-autoethnographic method, which as I read your response I feel is what you actually might have done aside from the content/thematic analysis - which would fit as the analytical tool that you used to approach your data.

Small point - on line 247 you still refer to "shades of grey" which is fine, but you might want to determine whether or not you want to keep it based on consistency.

Other comments regarding the response and revisions -

I wasn't suggesting you did or should perform a breaching experiment. Disaster scholars have long written on how these events (and a pandemic is very much a disaster) are akin to a breaching experiment en masse considering they reveal underlying social conditions and norms that left hidden in everyday practice. In the undoing of everyday life, disasters reveal what is left uncovered in everyday routines. Which you speak to in the conclusions. There is other important work by Martha Feldman with respect to routines as a source of change (2000) or other scholarship on the need to reproduce a sense of routine in times of disruption (e.g. Anthony Giddens and his concept of Ontological Security as written in the Constitution of Society (1991). This becomes especially important in your discussions about death, which I continue to think is fascinating.

You are almost there. I am pushing you because I really believe your work is important, especially in the downpour of COVID-related material that is just noise. This is especially true because it seems the author's major fields are medical and they are really bridging multiple disciplinary fields of sociology, anthropology, organizational studies, philosophy, in addition to methods, which is absolutely essential to deepening our understanding of how to deal with this pandemic in a more humanistic way.

Reviewer #4: Dear authors,

I have sent all of my detailed recommendations to the Editoral Office in order to contribute to the improvement of the manuscript.

Regards,

7. PLOS authors have the option to publish the peer review history of their article (what does this mean?). If published, this will include your full peer review and any attached files.

Reviewer #3: **Yes: **Natalie D Baker

Reviewer #4: No

---

## [Author Response · Author response to Decision Letter 2]

17 Jul 2021

Dear Ms Frey,

Please find attached our responses to the third and fourth reviewers. We hope that you will find this final revision in accordance to the reviewers’ requests.

Reviewer #3:

1. To clarify, I hope the authors understand this was a comment directed towards the prior reviewer and not the authors themselves.

2. I understand the authors points in response to my question and they are believable, especially since the authors define it as such:

"The preparedness period is represented by the phase when there is no direct contact with the threat, but all efforts are directed towards anticipating the course of events, preparing responses to those events and trying to translate those responses into practice. We believe that this period, which is characterized by a state of expectation, is one of the hardest to endure."

I am left with a couple important questions - 1) are the authors drawing on their own experiences too as they describe in their response letter? If so, then I suspect they have also engaged in a co-autoethnographic project, which adds even more legitimacy to this work (although is an often marginalized method). 2) how do they distinguish between the 'preparedness period' and the 'response period'? Perhaps, as authorities themselves of this context, these lines are blurry. Which in itself is extremely important as a finding and it worthy of mention. I am not trying to push more literature on to the authors or more work. However, I think 2 other papers that can help strengthen both the methods and findings, could be considered:

https://www.academia.edu/41122099/Hurricane_Harvey_Unstrapped_Experiencing_adaptive_tensions_on_the_edge_of_chaos

https://www.academia.edu/38022452/For_a_short_time_we_were_the_best_versions_of_ourselves_Hurricane_Harvey_and_the_Ideal_of_Community

The above works address the co-autoethnographic method, which as I read your response I feel is what you actually might have done aside from the content/thematic analysis - which would fit as the analytical tool that you used to approach your data.

Thank you for your comments and for your suggestion. I believe that the following phrase from our previous response to your commentaries may have been misleading „ Our experience, and also our colleagues’ experience, confirmed our feelings”. We read the papers which you recommended to us, and we have to be honest and to admit that we were not aware of this method. We are a team of doctors in internal medicine, and we are not familiarised with all the methods in qualitative research. We practice qualitative research out of passion, because we consider that in the medical field quantitative research is not enough, and that it must be associated with qualitative research in order to have an overview image, since medicine does not consists only in numbers, associations, causality, and p values, but it also involves humanism, which, unfortunately, in the era of randomised trials and big pharma, it had been overlooked. If it was to paraphrase Kahlil Gibran, we could say „In quantitative research or in qualitative research, the two side by side raise hands together to find what one cannot reach alone”. Our aim was not to test our feelings and assumptions through this study, but to discover how did our colleagues feel and how they experienced those rough moments. So, from a moral point of view, it would not be faire to say that we performed co-autoetnography since we were not even aware of this method. Moreover, the only moment when we let our experience infer in the results, was when we refered to the importance of keep working in the same team, since this issue was adressed during the interviews by a minority of participants, and we decided to include it in the results, due to the fact that we considered this important (which we mentioned in Statements section). Regarding the difference betweent the preparedness period and the response period, as we stated in the manuscript, the preparedness period is represented by the phase when there is no direct contact with the threat, while the response period would imply evaluating the experience and emotions of the healthcare staff while taking care of patients with COVID-19.

Small point - on line 247 you still refer to "shades of grey" which is fine, but you might want to determine whether or not you want to keep it based on consistency.

We rephrased that sentence.

Other comments regarding the response and revisions -

I wasn't suggesting you did or should perform a breaching experiment. Disaster scholars have long written on how these events (and a pandemic is very much a disaster) are akin to a breaching experiment en masse considering they reveal underlying social conditions and norms that left hidden in everyday practice. In the undoing of everyday life, disasters reveal what is left uncovered in everyday routines. Which you speak to in the conclusions. There is other important work by Martha Feldman with respect to routines as a source of change (2000) or other scholarship on the need to reproduce a sense of routine in times of disruption (e.g. Anthony Giddens and his concept of Ontological Security as written in the Constitution of Society (1991). This becomes especially important in your discussions about death, which I continue to think is fascinating.

You are almost there. I am pushing you because I really believe your work is important, especially in the downpour of COVID-related material that is just noise. This is especially true because it seems the author's major fields are medical and they are really bridging multiple disciplinary fields of sociology, anthropology, organizational studies, philosophy, in addition to methods, which is absolutely essential to deepening our understanding of how to deal with this pandemic in a more humanistic way.

Reviewer #4: Dear authors,

I have sent all of my detailed recommendations to the Editoral Office in order to contribute to the improvement of the manuscript.

Regards,

Please kindly find my recommendations for the manuscript below:

1. The content of the manuscript may have the potential to clarify the gray zones in pandemic due to the qualitative feature of the study.

We thank you for your appreciations. 

2. I had the sense that the manuscript is not limited to the preparedness phase as the content covers the actual and the active pandemic phases as well. The content should be revised by making it compatible with the aims of the manuscript.

We consider that the analysis presented in the results section is limited to the preparedness phase. Due to the fact that the virus spread to Romania later, weeks not only after the emergence of the virus, but also after the key moments when other countries were already engaged in an active fight against the virus, there are a number of references to particular situations experienced in other countries (for example the lack of personal protective equipments), which may give the feeling that the active pandemic phase is covered, but at the point when we performed the interviews, our hospital was preparing to become a COVID-19 unit, and none of the doctors have dealt with COVID-19 patients at that moment. As we stated in the introduction, the preparedness period „is represented by the phase when there is no direct contact with the threat, but all efforts are directed towards anticipating the course of events, preparing responses to those events and trying to translate those responses into practice”, and this particular issue is adressed in the results section. 

3. The authors are recommended to use another phrase instead of “bridge to heaven” as the subthemes does not fit with the title. In addition, it is not “death”, it is “life” which the health staff aims from the beginning to the end. As a last point, the authors are recommended to use a more objective description for the themes.

We believe that you have received the original version, which we submitted at first. This is the third time we are revising the manuscript, therefore major changes have been made until now, which we hope that will fulfill your requests. The themes have been modified previously according to the other three reviewers requests, and now the fourth theme „A bridge to heaven” does not have any subtheme. We believe that the title of the fourth theme is in accordance to its content; the theme ilustrates the fact that doctors had to hold on to what was familiar to them in order to be able to keep on track, and gives details about which were the particular things to help them achieve that. As we said, all the themes suffered major modifications, and we hope that you will find the actual content suitable. As you said, the aim of the healthcare personnel was life, but fear of death was present, an this was an important emotion which was heavily experienced at that point; so, it was mandatory to adress it in our results. Moreover, the third reviewer, which is an expert in social sciences, is particularly fond of this topic, and, as she said, considers it „fascinating”. Therefore, we hope that you will not be bothered by the fact that we have decided to keep the discussions about death. Regarding the last point you have mentioned, we tried to expose the results in an objective manner (which is more evident after the modifications we have made after the previous two major revisions); however, in our opinion, and according to Clarke and Brown, a certain degree of subjectivism is inherent to any qualitative research, and it differentiates it from quantitative research. 

4. The authors are recommended to add current/updated literature in their discussion section as there may be recent published studies.

We looked for recent qualitative studies to adress in the discussions section, regarding the way in which the healthcare staff experienced the preparedness period, but we did not identify any new studies regarding this topic. 

5. Limitations should be given in a separate and a more detailed content in the discussion part. The limitation of the study type should be emphasized more strictly.

We created a new section for Limitations, in which we further detailed what we consider to be limitations for our study. 

6. I could not understand how the authors will use their findings to improve the system. I suggest the authors to add detailed content especially into the discussion section.

We added in the Results section what could and should be done in order to improve the system, based on the results of our study. Unfortunately, it is not in our power to improve the system. Our aim was to gain a deeper understanding of how the healthcare staff experienced the preparedness period, which were the strengths and weaknesses in their vision, and to express them in order to gain awareness in this area. Ideas produce actions, and actions produce outcomes; therefore, after that, it is the duty of the healthcare system leaders (from macro to mycro) to implement measures like trainings for crisis situations, in order to achieve a level of preparedness that would permit doctors to face future pandemics without experiencing so acutely the fear of unknown, and the fear of death. 

7. Recommendations are suggested to be detailed.

It is not clear to what recommendations you are refering to; we suppose that you refer to the recommendations for improving the system, which we adressed in the previous section. 

8. The manuscript is recommended to be evaluated by a social scientist before giving the final decision.

The third reviewer that evaluated our paper is an expert in social sciences.

---

## [Decision Letter · Decision Letter 3]

9 Aug 2021

PONE-D-20-18668R3

COVID-19 pandemic preparedness period through healthcare workers’ eyes: a qualitative study from a Romanian healthcare facility

PLOS ONE

Dear Dr.Stoichitoiu ,

Thank you for submitting your manuscript to PLOS ONE. After careful consideration, we feel that it has merit but does not fully meet PLOS ONE’s publication criteria as it currently stands. Therefore, we invite you to submit a revised version of the manuscript that addresses the points raised during the review process.

Please address the minor revisions requested by Reviewer 4.

We look forward to receiving your revised manuscript.

Kind regards,

Rosemary Frey

Academic Editor

PLOS ONE

Journal Requirements:

Reviewers' comments:

Reviewer's Responses to Questions

**Comments to the Author**

1. If the authors have adequately addressed your comments raised in a previous round of review and you feel that this manuscript is now acceptable for publication, you may indicate that here to bypass the “Comments to the Author” section, enter your conflict of interest statement in the “Confidential to Editor” section, and submit your "Accept" recommendation.

Reviewer #3: All comments have been addressed

Reviewer #4: (No Response)

2. Is the manuscript technically sound, and do the data support the conclusions?

Reviewer #3: Yes

Reviewer #4: Yes

3. Has the statistical analysis been performed appropriately and rigorously? 

Reviewer #3: N/A

Reviewer #4: N/A

4. Have the authors made all data underlying the findings in their manuscript fully available?

Reviewer #3: Yes

Reviewer #4: Yes

5. Is the manuscript presented in an intelligible fashion and written in standard English?

Reviewer #3: Yes

Reviewer #4: Yes

6. Review Comments to the Author

Reviewer #3: I still hold that what you did was autoethnography to an extent despite not having knowledge of this method. That is not really a necessary condition, per se. And I admire that the research team chose to take on this form of methodology despite status as medical doctors as I would agree that it is extremely important for such studies to be conducted in these types of contexts.

Reviewer #4: Dear authors,

Review document about the manuscript has been prepared and sent to the Editorial Office.

Best Regards.

7. PLOS authors have the option to publish the peer review history of their article (what does this mean?). If published, this will include your full peer review and any attached files.

Reviewer #3: No

Reviewer #4: No

---

## [Author Response · Author response to Decision Letter 3]

11 Aug 2021

1. I could not see the “overall” recommendations as a separate part of the manuscript.

 In the previous recommendations, you suggested us the following ”Recommendations are suggested to be detailed”. We assumed that the recommendations you are referring to are in regard to the recommendations for improving the system. We modified at that time our recommendations, detailed them and they are presented in the end of our manuscript, where they are easy to be observed. However, we consider that presenting the recommendations in a separate part of the manuscript (as we did with the limitations of our study, according to your suggestion) would contravene to the general structure of an original research paper. 

2. The authors stated that their study was the first study. However, there may be

similar studies and the authors could not have accessed. They should change their sentence.

 At the moment when we submitted the original research article, more than one year ago, our study was the first study which addressed the preparedness phase of the COVID-19 pandemic. At your suggestion, we searched for recent qualitative studies regarding the way in which the healthcare staff experienced the preparedness period, but we did not identify any new studies regarding this topic. However, we changed our statement, adding the fact ”as far as we know” our study is the first study addressing this problem, in order to cover the studies which may have been published, but that are not available on PubMed.

3. It is still not clear how the authors have connected the “Bridge to heaven” part with

evidence-based medicine. 

 Given the fact that this study is a qualitative one in its nature, a sort of subjectivism is inherent to this design. However, we linked evidence-based medicine (if we can name evidence-based medicine the informations which were available at that time - they were mostly statements of different societies) to the title of the theme ”A bridge to heaven”, because, after we immersed into the data obtained from our participants, we considered that this emerged from it. As we tried to illustrate with our quotes, it was clear that for the healthcare staff any piece of trusted information made them feel more secure, gave them courage and energy to fully engage in this fight with the unknown, and together with other measures, we considered that this too is a bridge to heaven, heaven being here a metaphore for their mental well-being, which was absolutely mandatory in those times. 

4. As it is a qualitative research, social scientist’s evaluations should be officially

received before giving the decision.

 As far as we know, the 3rd reviewer, which choosed to make their name fully available for the peer-review process, is an expert in social sciences. 

Note. No need to send the revised manuscript to me, Editorial Office can give the final

decision especially using the social scientist's decision

---

## [Decision Letter · Decision Letter 4]

1 Sep 2021

COVID-19 pandemic preparedness period through healthcare workers’ eyes: a qualitative study from a Romanian healthcare facility

PONE-D-20-18668R4

Dear Dr. Stoichitoiu,

We’re pleased to inform you that your manuscript has been judged scientifically suitable for publication and will be formally accepted for publication once it meets all outstanding technical requirements.

Kind regards,

Rosemary Frey

Academic Editor

PLOS ONE

Additional Editor Comments (optional):

Reviewers' comments:

Reviewer's Responses to Questions

**Comments to the Author**

1. If the authors have adequately addressed your comments raised in a previous round of review and you feel that this manuscript is now acceptable for publication, you may indicate that here to bypass the “Comments to the Author” section, enter your conflict of interest statement in the “Confidential to Editor” section, and submit your "Accept" recommendation.

Reviewer #3: All comments have been addressed

Reviewer #4: All comments have been addressed

2. Is the manuscript technically sound, and do the data support the conclusions?

Reviewer #3: Yes

Reviewer #4: Yes

3. Has the statistical analysis been performed appropriately and rigorously? 

Reviewer #3: N/A

Reviewer #4: N/A

4. Have the authors made all data underlying the findings in their manuscript fully available?

Reviewer #3: Yes

Reviewer #4: Yes

5. Is the manuscript presented in an intelligible fashion and written in standard English?

Reviewer #3: Yes

Reviewer #4: Yes

6. Review Comments to the Author

Reviewer #3: Endorse publication. I have already endorsed in the prior review and I have no additional comments to add.

Reviewer #4: Dear authors,

I have read the revised manuscript.

Thank you for covering almost all of my recommendations.

Regards,

7. PLOS authors have the option to publish the peer review history of their article (what does this mean?). If published, this will include your full peer review and any attached files.

Reviewer #3: No

Reviewer #4: No

---

## [Editor Report · Acceptance letter]

28 Sep 2021

PONE-D-20-18668R4 

COVID-19 pandemic preparedness period through healthcare workers’ eyes: a qualitative study from a Romanian healthcare facility 

Dear Dr. Stoichitoiu:

I'm pleased to inform you that your manuscript has been deemed suitable for publication in PLOS ONE. Congratulations! Your manuscript is now with our production department. 

Kind regards, 

on behalf of

Dr. Rosemary Frey 

Academic Editor

PLOS ONE